# Obesity Does Not Influence Delayed Gastric Emptying Following Pancreatoduodenectomy

**DOI:** 10.3390/biology11050763

**Published:** 2022-05-17

**Authors:** Jana Enderes, Christiane Pillny, Hanno Matthaei, Steffen Manekeller, Jörg C. Kalff, Tim R. Glowka

**Affiliations:** Department of Surgery, University Hospital Bonn, 53127 Bonn, Germany; jana.enderes@ukbonn.de (J.E.); christiane.pillny@ukbonn.de (C.P.); hanno.matthaei@ukbonn.de (H.M.); steffen.manekeller@ukbonn.de (S.M.); kalff@uni-bonn.de (J.C.K.)

**Keywords:** pancreaticoduodenectomy, whipple, obesity, body mass index, delayed gastric emptying

## Abstract

**Simple Summary:**

Over the last decades the number of obese patients has been increasing. Not only is obesity associated with other diseases such as cardiovascular diseases, diabetes and asthma but obese patients are also at a higher risk for developing different types of cancers, for instance pancreatic cancer with a consecutive increased need for pancreatic surgery. Even though it is not life threatening, impaired gastric motility, also known as delayed gastric emptying, has still remained the most common complication after pancreatic surgery. However, the data about obesity on postoperative outcome after pancreatic surgery are inconsistent, specifically in relation to delayed gastric emptying. The goal of this study was to investigate the impact of obesity on postoperative outcome, specifically on delayed gastric emptying, after pancreatic surgery. Our data show no difference in the occurrence and severity of delayed gastric emptying in patients with obesity compared to non-obese patients. Moreover, the overall mortality rate did not differ between the two groups. In summary, our data show that obese patients are not put at a higher risk in regard to postoperative outcome, which makes pancreatic surgery a feasible procedure in the obese patient, specifically in relation to delayed gastric emptying.

**Abstract:**

Background: The data about obesity on postoperative outcome after pancreatoduodenectomy (PD) are inconsistent, specifically in relation to gastric motility and delayed gastric emptying (DGE). Methods: Two hundred and eleven patients were included in the study and patients were retrospectively analyzed in respect to pre-existing obesity (obese patients having a body mass index (BMI) ≥ 30 kg/m^2^ vs. non-obese patients having a BMI < 30 kg/m^2^, *n* = 34, 16% vs. *n* = 177, 84%) in relation to demographic factors, comorbidities, intraoperative characteristics, mortality and postoperative complications with special emphasis on DGE. Results: Obese patients were more likely to develop clinically relevant pancreatic fistula grade B/C (*p* = 0.008) and intraabdominal abscess formations (*p* = 0.017). However, clinically relevant DGE grade B/C did not differ (*p* = 0.231) and, specifically, first day of solid food intake (*p* = 0.195), duration of intraoperative administered nasogastric tube (NGT) (*p* = 0.708), rate of re-insertion of NGT (0.123), total length of NGT (*p* = 0.471) or the need for parenteral nutrition (*p* = 0.815) were equally distributed. Moreover, mortality (*p* = 1.000) did not differ between the two groups. Conclusions: Obese patients do not show a higher mortality rate and are not at higher risk to develop DGE. We thus show that in our study, PD is feasible in the obese patient in regard to postoperative outcome with special emphasis on DGE.

## 1. Introduction

Obesity is defined as abnormal or excessive fat accumulation in the human body that impairs health and it is assessed by body mass index (BMI, defined as the weight in kilograms divided by height in meters squared) with a BMI ≥ 30 kg/m^2^ [1]. Over the last decades obesity has been increasing and has become a tremendous health burden. In 2015, obesity was seen in more than 600 million adults worldwide and its prevalence has doubled since 1980 [2,3].

Obesity is known to have a tremendous impact on health since it is related to numerous comorbidities such as cardiovascular diseases, diabetes mellitus, chronic kidney diseases, musculoskeletal disorders and different types of cancer [2]. In particular, obesity is also associated with pancreatic diseases such as acute pancreatitis and pancreatic cancer [4,5,6]. The latter, with pancreatic ductal adenocarcinoma (PDAC), is recognized as being one of the most aggressive solid tumors and is predicted to be the second cause of cancer related-deaths in 2030 [7]. The only potential curative therapy for PDAC is surgical resection followed by adjuvant therapy for locally resectable tumors or for borderline resectable tumors after neoadjuvant therapy [8]. 

Thus, since the prevalence of obesity has been evolving over the last years, with obesity being known to be associated with pancreatic cancer, there will be an increased number of obese patients with the need for pancreatic surgery. However, the data on the impact of obesity on postoperative outcome, specifically on delayed gastric emptying (DGE) as the most common complication following pancreatoduodenectomy [9,10], are controversial. Whereas the majority of studies investigating general risk factors for developing DGE report a correlation between BMI and a higher risk for DGE [11,12,13,14], there are some studies reporting no difference in the occurrence of DGE [15,16,17]. Moreover, studies that concentrated on postoperative outcomes after PD in patients with a higher BMI are controversial [18,19,20,21,22,23,24,25]. While some studies put obese patients at higher risk for developing DGE [20,24,25], others do not report a difference in the occurrence of DGE [19,22] and again others did not even focus on investigating DGE [18,23]. 

Based on our own observations we can conclude that obesity is not necessarily associated with more severe DGE. Due to these controversial results, and in order to validate them, we decided to analyze our own results and investigate the correlation between obesity and outcome after pancreatoduodenectomy with special emphasis on DGE including specific parameters according to the definitions of the International Study Group of Pancreatic Surgery (ISGPS), since none of the mentioned studies used these.

## 2. Materials and Methods

Between January 2015 and May 2021, 232 patients had surgery for pancreatic head resection at our center and were retrospectively analyzed from our prospectively recorded pancreatic resection database. We proceeded after having obtained written informed consent from the patients and with the approval of the institutional ethics committee (ethics committee of the Rheinische Friedrich-Wilhelms University of Bonn, 347/13). We excluded patients who showed a history of gastric resection (*n* = 2) and patients with fasting periods not associated with DGE, such as long-term ventilation > 7 days, dysphagia and fasting due to octreotide therapy during the first five days after surgery for high-risk soft pancreatic tissue or due to endoscopic vacuum-assisted closure therapy for clinically relevant PF (*n* = 19) and thus 211 patients were included in the study. Two groups were established according to standard definitions of the World Health Organization (WHO) [1]: the obese group comprised 34 patients with a BMI ≥ 30 kg/m^2^, including class 1 obesity (BMI ≥ 30 < 35 kg/m^2^, *n* = 26), grade 2 obesity (BMI ≥ 35 < 40 kg/m^2^, *n* = 4) and class 3 obesity (BMI ≥ 40 kg/m^2^, *n* = 4). The non-obese group comprised 177 patients with having each a BMI < 25 kg/m^2^. BMI was then analyzed with a special emphasis on DGE and DGE-related parameters. These parameters include the first day on which the patient starts to eat solid food, the length of the nasogastric tube (NGT) that is administered during surgery by default, the need for reinsertion of an NGT once the intraoperative-administered tube was removed, the total length of an inserted NGT (total length of intraoperatively-administered tube and, in case of reinsertion, this includes the total length of the reinserted tube) as well as the need for parenteral nutrition. Postoperative complications were documented according to the Clavien–Dindo classification [26] and the International Study Group on Pancreatic Surgery (ISGPS) definitions were used to classify the degree of specific complications after pancreatic resection such as pancreatic fistula (PF), postpancreatectomy hemorrhage (PPH) and DGE [9,27,28]. DGE is defined as follows: grade A DGE occurs if an NGT is required between postoperative day (POD) 4 and 7 or in case of reinsertion of NGT after POD 3; grade B DGE occurs if NGT is required between POD 8 and 14 or if reinsertion of NGT is necessary after POD 7; grade C DGE occurs if NGT is required after POD 14 or in case of reinsertion of NGT after POD 14. Additionally, further parameters such as change in clinical management and the need for prokinetic drugs, parenteral nutrition and interventional treatment can be taken in to account to define DGE more precisely: with grade A DGE not leading to a major change in clinical management and not prolonging the total length of the hospital stay; grade B DGE leading to an adjustment of the clinical management including administration of prokinetic drugs and parenteral nutrition and the need for reinsertion of a gastric tube prolonging the total length of the hospital stay; grade C DGE leading to a major change in clinical management with a need for parenteral nutrition over 3 weeks and interventional treatment of associated complications such as abscess drainage or relaparotomy also prolonging the total length of the hospital stay [9]. The ISGPS definitions of DGE have been extensively reevaluated and been found feasible in diagnosing DGE [29].

Perioperative management was carried out according to our institutional standard operating procedure protocol. This includes discussing the patient in our multidisciplinary Tumor Board, not performing bowel preparations, providing supplementary feeding if the patient is malnourished and allowing the patient to eat and drink six and two hours prior to the procedure, respectively. 

PD was performed by 3 certified senior pancreatic surgeons (JCK, SM, TRG). The procedures were performed standardized with a single-loop technique either ante- or retrocolically and in the latter case, with supra- or infracolic routes [30]. Duodenoenterostomy, pancreatogastrostomy and end-to-side choledochojejunostomy were performed as previously described [31,32]. In case of infiltration of the antrum, a classic Whipple with double-loop reconstruction was performed. Standardized or extended lymphadenectomy defined by the ISGPS definitions [33] was performed and a sample of the gall fluid was taken by default and sent for microbiological analysis.

Perioperatively, in order to prevent postoperative pain, patients received either a peridural catheter or opioids in a patient-controlled manner. A 14 French nasogastric tube (NGT) and two soft drains were placed by default, the latter at the sites of pancreatogastrostomy and choledochojejunostomy before closure of the abdomen.

Postoperatively, patients spent a minimum of one night in the intensive care unit. Directly after surgery patients were allowed to drink water. Diet started on postoperative day (POD)3 if amylase levels in the drainage fluid were normal. In this case, soft drains were removed and the diet was started with easily digestible/fat-reduced meals and easily digestible/fiber-reduced meals on POD4. On POD5 patients received a basic diet (no pulses/no brassica) and a normal diet on POD6. If daily secretions were less than 500 mL NGT was removed and if patients showed signs of vomiting, transition to a normal diet was discontinued and an NGT was re-inserted. Octreotide (100 μg 3×/d s.c. for 5 days) was given in case of PF, as a laxative magnesium sulfate was given on POD2. All patients received an antibiotic prophylaxis with an aminopenicillin plus β-lactamase inhibitor and a weight adapted thrombosis prophylaxis. 

Data were recorded and analyzed with Excel 2013 (Microsoft Corporation, Redmond, Washington, DC, USA) and SPSS 24 (IBM Corporation, Armonk, New York, NY, USA). Continuously and normally distributed variables were expressed as medians ± standard deviation and analyzed using the Student’s *t*-test. Non-normally distributed data were expressed as medians and interquartile range and analyzed using the Mann–Whitney U test. Categorical data were expressed as proportions and compared with the Pearson x^2^ or the Fisher’s exact test. Factors with a *p*-value < 0.1 in the univariate analysis were included in the multivariate stepwise logistic regression with a significance level of *p* < 0.05 for inclusion and *p* < 0.10 for removal 95 in each step. The relative risk was described by the estimated odds ratio with 95% confidence intervals. A *p*-value < 0.05 was considered statistically significant [32].

## 3. Results

Regarding demographic characteristics, patients with obesity were of the same age than non-obese patients (67 years vs. 68 years, *p* = 0.374) and they were mainly female (59% female vs. 41% male, *p* = 0.037). Preoperative conditions such as alcohol abuse, nicotine consumption and weight loss did not differ between the two groups. The comorbidities included and measured by the Charlson Morbidity Index (CCI), as well as the perioperative risk represented by the American Society of Anesthesiologists (ASA)-score were also equally distributed amongst them (Table 1). However, obese patients showed a higher existence of preoperative diabetes mellitus (44% vs. 27%, *p* = 0.047) and they preoperatively received less biliary drainage (26% vs. 50%, *p* = 0.011) even though cholangitis occurred equally in both groups (3% vs. 10%, *p* = 0.323). Intraoperative data such as the duration of the surgical procedure, blood loss and subsequent transfusion of erythrocyte concentrates were equal in both groups (Table 1). However, obese patients showed less positive intraoperatively-taken microbiological gall fluid cultures (26% vs. 59%, *p* = 0.003). Tumor and organ related characteristics, such as tumor size, pancreatic texture and dilated pancreatic duct, were also equally distributed in obese and non-obese patients (Table 1). Regarding technical aspects, particularly, we did not observe a difference in pylorus-preserving or classic Whipple procedure nor in ante- or retrocolic and if the latter was chosen infra- or supracolic reconstructions between obese and non-obese patients. Moreover, in obese as well as non-obese patients extended lymphadenectomy was performed equally (53% vs. 58%, *p* = 0.613) and the amount of resected lymph nodes were the same in both groups (24 (15–27) vs. 24 (17–32, *p* = 0.252)). Postoperatively, the groups were comparable regarding the duration of the in hospital stay as well as the stay in the intensive care unit (Table 1). 

As for postoperative complications, both groups were comparable regarding major postoperative complications (Clavien III–IV) and mortality (Table 2). In addition, both groups showed a comparable rate of insufficiencies of BDA or DE and the need for a second surgery did not differ between the two groups (reasons for reoperation are given in Table 3). Obese patients did not show a higher rate of suprafascial wound infections (21% vs. 18%, *p* = 0.741), however, interestingly, we did observe significantly more intraabdominal infections with intraabdominal abscess formations in patients with pre-existing obesity (29% vs. 13%, *p* = 0.017). Moreover, in the obese population we observed significantly more clinically relevant pancreatic fistula grade B/C (35% vs. 16%, *p* = 0.008), which were not automatically accompanied by prolonged fasting times. 

Obese and non-obese patients showed comparable occurrence of PPH and DGE, and notably, the specific DGE-related parameters as shown in Table 4 were comparable between obese and non-obese patients (Table 4).

Univariate analysis for predictors of DGE such as single or double loop, retro- or antecolic and infra- or supracolic reconstructions, pancreatic texture, pancreatic duct size, as well as PF, PPH and intraabdominal abscess formation did not reveal a correlation between DGE and these parameters. In particular, obesity was not a risk factor for DGE. When analyzing predictors for an increased mortality, the following parameters qualified for multivariate analysis: reoperation, pancreatic fistula grade C, insufficiency of BDA, insufficiency of DE and reinsertion of gastric tube (Table 5). In the multivariate analysis, only reoperation was a risk factor for high mortality (*p* ≤ 0.001).

## 4. Discussion

Preoperative conditions, such as comorbidities in general as well as the perioperative risk of obese and non-obese patients, did not differ between the two groups. As expected, diabetes mellitus was more often seen in obese patients. This is not surprising since obesity is known to be associated with type 1 and type 2 diabetes mellitus, of which the latter is accompanied with decreased insulin production and peripheral insulin resistance subsequently leading to hyperglycemia [34]. Even in patients not having been diagnosed with diabetes prior to PD, decreased insulin levels and hyperglycemia have been observed due to a reduction in insulin-producing β-cells. Moreover, secretion of GLP-1, a hormone that is secreted after meal ingestion and stimulates insulin secretion, has been observed after PD, with both, GLP-1 and hyperglycemia being known to delay gastric emptying [35]. Thus, one would assume that, after PD, obese patients show even more decreased gastric motility and suffer from DGE more often, especially since an increased amount of GLP-1 has been observed in obese patients [36]. However, due to its incretin function, GLP-1 is also known to normalize plasma glucose, especially in patients suffering from obesity and diabetes which might act as a counter mechanism for the increased hyperglycemia-induced impaired gastric motility observed after PD [36,37]. Knowing that the occurrence of DGE did not differ between non-obese and obese patients and thus knowing that the latter do not necessarily face possible consequences that might be associated with DGE such as prolonged hospital stay, delay of adjuvant therapy and impaired cancer-specific survival [38] is an important finding. 

Our findings are in line with other studies investigating general risk factors associated with DGE [15,17], though, there are also other studies that did identify an association between a high BMI and the development of DGE [11,12,13,14]. Additionally, studies that investigated the general outcome in obese patients after PD are inconclusive, with some studies putting obese patients at a higher risk to develop DGE [20,24,25], whereas others did not report a higher occurrence of DGE in obese patients [16,19,22] and even other studies did not consider DGE as an important outcome parameter after PD at all [18,23]. However, the comparability between these studies is very limited, especially since different cut-off points of BMI and different definitions of obesity are being used. Indeed, Shamali et al. and Chang et al. chose the same BMI cut off points than we did, distinguishing between obese patients and non-obese patients [22,23], though the latter mentioned study did not investigate specific complications after PD such as the occurrence of DGE at all. Shamali et al. found out that obese patients are not at a higher risk to develop DGE after PD, which is in line with our findings, however, as most of the studies that did investigate the correlation between obesity and DGE, this study also lacks ISGPS definitions [19,20,22]. Nevertheless, in our study we used WHO-classified definitions of obesity and to the best of our knowledge, this is the first study investigating the effect of DGE according to ISGPS definitions, specifically by using a sub-classification into grade A, B and C as well as special parameters such as first day of solid food intake, duration of intraoperative administered NGT, rate of re-insertion of an NGT or the need for parenteral nutrition. Data on the frequency of DGE following minimally invasive pancreatic surgery is contradictory [39,40,41]. A meta-analysis of existing cohort and register studies showed less DGE in the minimally invasive group [42]. Since only 16% of our patients were resected robotically, we cannot say if this has an effect on DGE in our study.

In our study, obese patients preoperatively received less biliary drainage even though cholangitis occurred equally in both groups. Whether this is due to a hesitation in using preoperative biliary drainage because of a higher fear of complications after endoscopic retrograde cholangiopancreatography (ERCP) even though obesity does not seem to be associated with a higher incidence of complications, such as post-ERCP pancreatitis [43], remains an open question. Moreover, obese patients showed less positive intraoperatively-taken bile cultures, assumingly since they preoperatively received less biliary drainage which is known to lead to a loss of the antimicrobial defense due to sphincter oddi dysfunction and subsequent bacterial colonization of the bile duct [44]. 

Positive bile duct cultures after PD are associated with an increased risk for postoperative infectious complications such as intraabdominal abscess formation and wound infections [45,46], the latter known to be a common complication seen in the obese population after PD [23,24]. In our study, obese patients did not show more wound infections, which therefore could be explained by the less positive bile duct cultures. However, even though obese patients showed positive bile duct cultures less often, they still developed intraabdominal abscess formations more often. This is not surprising, because the impaired immune function seen in obese patients might explain this observation [47], furthermore and far more importantly, one of the main complications seen in the obese population after PD is the development of pancreatic fistulas [19,20,22,24,25,48] that can lead to intraabdominal abscess formations. Indeed, in this study, obese patients showed more often clinically relevant pancreatic fistulas and developed more often intraabdominal abscess formations. Importantly, this did not result in a higher rate of reoperations. Intraabdominal abscesses were treated interventionally by either CT-guided drainage or endoscopic transgastric drainage and this did not affect the length of the hospital stay, which was 23 days in the obese group and 22 days in the non-obese group. The length of the hospital stay not only reflects the degree of postoperative complications [49] but it is also influenced by regional and cultural differences. Whereas studies conducted in the United States are known to report rather a short length of hospital stay [49], studies conducted in Japan usually report longer hospital stays up to 47.7 days [50]. In Germany, the mean hospital stay according to the diagnosis related group classification system for PDs is 17 days. The slightly increased length of hospital stay observed in this study is in line with the observed rather late first day of solid food intake on POD 12 and 10, respectively. DGE is known to increase hospital stay [51], however conclusions need to be drawn carefully here since first day of solid food intake is not the only parameter defining DGE. According to the ISGPS definitions, DGE is defined by the duration of NGT requirement and NGT reinsertion, and additionally by further parameters such as change in clinical management and the need for prokinetic drugs, parenteral nutrition and interventional treatment. Especially since DGE rates in this study are only 32% and 23%, respectively, we thus believe that not only DGE but also other factors such as a strong patient’s will to stick to light food or a non-adherence to our enhanced recovery program are accountable for prolonging first day of solid food intake and thus leading to a longer hospital stay. Furthermore, the overall mortality rate was not affected but in case of reoperation, patients showed a higher mortality rate which is in line with our previous studies after PD [52] or after liver transplantation [53].

Since this is a retrospective study, it has several limitations including data interpretation. This study was single-centered with a small sample size including only 211 patients. Therefore, further multicenter- or register-based studies are needed to validate the present results before definitive conclusions can be drawn with respect to the small sample size within the obese group.

## 5. Conclusions

In summary, to the best of our knowledge this is the first study investigating the effect of WHO-classified obesity on delayed gastric emptying according to ISGPS definitions, particularly by using sub-classifications into grade A, B and C as well as special parameters such as first day of solid food intake, duration of intraoperative-administered NGT, rate of re-insertion of an NGT or the need for parenteral nutrition. We show that, according to the ISGPS definitions, obese patients are not at a higher risk of developing DGE. Obesity was associated with increased risks of pancreatic fistulas and intraabdominal abscess formations; however, neither of these affected the need for reoperation nor the overall mortality. Taken together, we show that obese patients are not at higher risk of developing DGE which makes PD a feasible procedure in the obese patient in regard to postoperative outcome with special emphasis on DGE.

## Figures and Tables

**Table 1 biology-11-00763-t001:** Demographic and perioperative data.

	Obese	None-Obese	*p*
	*n* = 34	*n* = 177	
Age (a)	67 (60–72)	68 (59–76)	0.374
Gender female	20 (59%)	70 (40%)	0.037
Diagnosis malignant	26 (76%)	135 (76%)	0.980
Alcohol abuse	5 (15%)	47 (27%)	0.122
Nicotine (active consumption)	8 (24%)	53 (30%)	0.395
Weight loss	20 (59%)	100 (56%)	0.912
Charlson Comorbidity Index	3 (2–3)	3 (2–4)	0.885
ASA physical status classification	2 (2–3)	3 (2–3)	0.323
Cholangitis	1 (3%)	18 (10%)	0.323
Preoperative biliary drainage	9 (26%)	89 (50%)	0.011
Preoperative diabetes mellitus	15 (44%)	48 (27%)	0.047
Duration of operation (min)	413 (306–492)	372 (311–442)	0.301
Blood loss (mL)	500 (275–925)	600 (350–1000)	0.214
Transfusions (erythrocyte concentrate)	0 (0)	0 (0–2)	0.086
Positive intraoperative microbiology	9 (26%)	104 (59%)	0.003
Tumor size (cm)	3 (2–4)	3 (2–4)	0.524
Soft pancreas parenchyma	18 (53%)	76 (43%)	0.189
Pancreatic duct > 5 mm	5 (15%)	41 (23%)	0.395
Extended lymphadenectomy	18 (53%)	102 (58%)	0.613
Resected lymph nodes	24 (15–27)	24 (17–32)	0.252
Pylorus-preserving procedure	31 (91%)	153 (86%)	0.582
Retrocolic duodenoenterostomy	29 (85%)	154 (87%)	0.779
Infracolic reconstruction	11 (32%)	69 (39%)	0.550
Robotic operation	8 (24%)	25 (14%)	0.176
Stay in hospital (d)	23 (18–30)	22 (17–31)	0.851
Stay in intensive care unit (d)	1 (1–3)	2 (1–3)	0.538
Stay in intensive care unit with respirator (d)	0 (0)	0 (0)	0.200

Data are shown as frequency (%) or median (interquartile range); ASA, American Society of Anesthesiologists.

**Table 2 biology-11-00763-t002:** Postoperative outcome/complications.

	Obese	Non-Obese	*p*
	*n* = 34	*n* = 177	
Clavien major (grade III-IV)	18 (53%)	85 (48%)	0.599
Mortality	1 (3%)	6 (3%)	1.000
Insufficiency of BDA	3 (9%)	8 (5%)	0.390
Insufficiency of DE	0 (0%)	6 (3%)	0.592
Reoperation	4 (12%)	17 (10%)	0.754
Wound infection (suprafascial)	7 (21%)	32 (18%)	0.741
Intraabdominal abscess formation	10 (29%)	23 (13%)	0.017
PF grade B/C	12 (35%)	28 (16%)	0.008
PPH grade B/C	11 (32%)	44 (25%)	0.362
DGE grade B/C	11 (32%)	40 (23%)	0.231

Data are shown as frequency (%); BDA, biliodigestive anastomosis; DE, duodenoenterostomy; PF, pancreatic fistula; PPH, postpancreatectomy hemorrhage; DGE, delayed gastric emptying.

**Table 3 biology-11-00763-t003:** Reasons for reoperation.

Obese	(*n* = 4)	n	
		2	Early insufficiency of BDA
		1	Bleeding at PG site
		1	Insufficiency of PG
**Non-Obese**	**(*n* = 17)**	**n**	
		5	Early insufficiency of BDA
		3	Insufficiency of PG
		2	Pancreatitis
		2	SSI
		1	Intraabdominal bleeding (A. hepatica)
		1	Insufficiency of GE
		1	Insufficiency gastrotomy
		1	Necrosis of spleen
		1	Ischemia right hemicolon

Data are expressed as numbers; BDA, biliodigestive anastomosis; PG, pancreatogastrostomy; SSI, surgical site infection.

**Table 4 biology-11-00763-t004:** Delayed gastric emptying—parameters according to ISGPS.

	Obese	Non-Obese	*p*
	*n* = 34	*n* = 177	
First day of solid food intake	12 (8–17)	10 (7–15)	0.195
Intraoperative administered nasogastric tube (d)	4 (3–6)	4 (3–7)	0.708
Reinsertion of gastric tube	15 (44%)	49 (28%)	0.123
Total length of inserted nasogastric tube * (d)	9 (6–11)	7 (5–10)	0.471
Parenteral nutrition (d)	4 (0–7)	3 (0–7)	0.815

Data are shown as frequency (%) or median (interquartile range); * comprises the total length of the intraoperatively administered tube and, in case of reinsertion, includes the total length of the reinserted tube.

**Table 5 biology-11-00763-t005:** Risk factors associated with high mortality.

	Odds Ratio	95%-CI	*p*
**Univariate**			
Reoperation	29.219	5.245–162.758	≤0.001
PF grade C	29.850	5.238–170.107	0.001
Insufficiency of BDA	8.667	1.475–50.919	0.045
Insufficiency of DE	19.900	2.931–135.112	0.013
Reinsertion of gastric tube	4.300	0.766–24.129	0.092
BMI ≥ 30 kg/m^2^	0.864	0.101–7.411	1.000
**Multivariate**			
Reoperation	23.067	3.900–136.436	≤0.001

BDA, biliodigestive anastomosis; BMI, body mass index; CI, confidence interval; DGE, delayed gastric emptying; PF, pancreatic fistula.

## Data Availability

Our anonymized pancreatic resection database contains sensible data (e.g., date of surgery), with which certain patients could be identified. According to German law and according to the approval of the ethics committee, these data must not be published. Access to the database can be obtained from the corresponding author upon reasonable request.

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
