# Peer review of "Obesity Does Not Influence Delayed Gastric Emptying Following Pancreatoduodenectomy"

_biology, 2022, doi:10.3390/biology11050763_

Round 1

Reviewer 1 Report

Dr. Enderes et al. showed comparable incidence of DGE after pancreatoduodenectomy between obese and non-obese patients.

As the authors commented, incidence of DGE based on the grading defined by ISGPF has not been well evaluated compared with that of POPF. In this respect, the paper included valuable information. However, there are several concerns to be addressed as the below.

Major comments

  1. The main focus of this study is the comparison of the incidence of DGE between obese and non-obese patients. The focus is unclear in the Methods section. What kinds of outcomes were evaluated and how should be specifically described.
  2. The authors presented univariate and multivariate analyses of the predictors of mortality rather than DGE, and this is another reason that I feel the paper is unfocused. To be specific to DGE as the main outcome to be evaluated, univariate and multivariate analyses for the predictors of DGE should be presented including perioperative variables such as the route of duodenoenterostomy, POPF, and abdominal abscess.

Minor comments

  1. Because DGE has been less commonly evaluated compared with POPF, please briefly add the definition of each grade of DGE in introduction or methods section.
  2. Please spell out the abbreviation of "PPH" not only in the footnote of Table 2 but also in the text (in Methods section).
  3. Is end-to-end coledochojejunostomy true? I think end-to-side anastomosis is common. 
  4. Please present the reasons of reoperation.
  5. What is the definition of "solid food"? The authors described in the methods that patients were allowed to take normal diet on PDO6. On the other hand, Table 3 suggested most patients started solid food only after POD6.
  6. In Table 3, the term "intraoperative gastric tube (d)" is a bit confusing. Please explain the difference from "total length of inserted gastric tube (d)".

Reviewer 2 Report

Thank you for the retrospective single center cohort study investigating the impact of obesity on DGE. 

The manuscript  (MC) is well written and data presentation is fine. However the MC should be revised prior to publication:

  • first limitations should be named and discussed (e.g. retrospective design, sample size (16% vs. 84%), and so on) in the text; furthermore it should be mentioned that e.g. multicenter- or register-based studies are needed to validate the present results before definitive conclusions can be drawn with respect to the small sample size within the obese group
  • retrospective design should be mentioned in the abstract
  • authors should discuss the reasons for the long length of stay (23 and 22 days) and the late first day of solid food intake (POD 12 vs. 10) especially that  DGE is primary endpoint but the rate of DGE is only 32 and 23% within the groups

Minor:

  • Dindo-Clavien is not a commen term. it should be named Clavien-Dindo

Round 2

Reviewer 2 Report

Thank you for revisiong the MC.

Author Response

Dear Reviewer 2, sincere thanks for your valuable comments. These were really helpful!